

# Global evaluation and calibration of a passive air sampler for gaseous mercury

David S. McLagan[1], Carl P. J. Mitchell[1], Alexandra Steffen[2], Hayley Hung[2], Cecilia Shin[2], Geoff W. Stupple[2], Mark L. Olson[3], Winston T. Luke[4], Paul Kelley[4], Dean Howard[5], Grant C. Edwards[5], Peter F. Nelson[5], Hang Xiao[6], Guey-Rong Sheu[7], Annekatrin Dreyer[8], Haiyong Huang[1], Batual Abdul Hussain[1], Ying D. Lei[1], Ilana Tavshunsky[1], and Frank Wania[1]

[1] Department of Physical and Environmental Sciences, University of Toronto Scarborough, Toronto, M1C 1A4, Canada

[2] Air Quality Processes Research Section, Environment and Climate Change Canada, Toronto, M3H 5T4, Canada

[3] Atmospheric Mercury Network, National Atmospheric Deposition Network, Champaign, 61820-7495, USA

[4] Air Resources Lab, National Oceanic and Atmospheric Administration, Maryland, 20740, USA

[5] Department of Environmental Sciences, Macquarie University, Sydney, 2109, Australia

[6] Center for Excellence in Regional Atmospheric Environment, Institute of Urban Environment, Xiamen, 361021, China

[7] Department of Atmospheric Sciences, National Central University, Taoyuan City, 32001, Taiwan

[8] Air Monitoring, Eurofins GfA, Hamburg, 21107, Germany

*Correspondence to:* Frank Wania (frank.wania@utoronto.ca)





**Abstract.** Passive air samplers (PASs) for gaseous mercury (Hg) were deployed for time periods between 1 month and 1 year at 20 sites across the globe with continuous atmospheric Hg monitoring using active Tekran instruments. The purpose was to evaluate the accuracy of the PAS vis-à-vis the industry standard active instruments and to determine a sampling rate ($SR$; the

volume of air stripped of gaseous Hg per unit of time) that is applicable across a broad variety of conditions. The sites spanned a wide range of latitudes, altitudes, meteorological conditions, and gaseous Hg concentrations. Precision, based on 378 replicated deployments performed by numerous personnel at multiple sites, is $3._6 \pm 3._0$ %[*], confirming the PAS's excellent reproducibility and ease-of-use. Using a $SR$ previously determined at a single site, gaseous Hg

concentrations derived from the globally distributed PASs deviate from Tekran-based concentrations by $14._2 \pm 10$ %. A recalibration using the entire new data set yields a slightly higher $SR$ of $0.135_4 \pm 0.016$ m$^3$ day$^{-1}$. When concentrations are derived from the PAS using this revised $SR$ the difference is reduced to $8._8 \pm 7._5$ %. At the mean gaseous Hg concentration across the study sites of 1.54 ng m$^{-3}$, this represents an ability to resolve concentrations to within 0.13

ng·m$^{-3}$. Adjusting the sampling rate to deployment specific temperatures and wind speeds does not decrease the difference in active–passive concentration further ($8._7 \pm 5._7$ %), but reduces its variability by leading to better agreement in Hg concentrations measured at sites with very high and very low temperatures and very high wind speeds. This value ($8._7 \pm 5._7$ %) represents a conservative assessment of the overall uncertainty of the PAS due to inherent uncertainties of the

Tekran instruments. Going forward, the recalibrated $SR$ adjusted for temperature and wind speed should be used, especially if conditions are highly variable or deviate considerably from the average of the deployments in this study ($9.8_9$ °C, $3.4_1$ m s$^{-1}$). Overall, the study demonstrates that the sampler is capable of recording background gaseous Hg concentrations across a wide range of environmental conditions with accuracy similar to that of industry standard active sampling

instruments. Results at sites with active speciation units were inconclusive on whether the PASs take up total gaseous Hg or solely gaseous elemental Hg primarily because gaseous oxidized Hg concentrations were in a similar range as the uncertainty of the PAS.

---

[*] subscripted numbers are not significant, but are reported to reduce rounding errors in subsequent studies (see Section 2.3 for details)



# 1. Introduction

Article 19 of the Minamata Convention requests that "parties endeavour to cooperate to develop
and improve … geographically representative monitoring of mercury (Hg) levels in
environmental media … [and gain] … information on the environmental cycle, transport
(including long-range transport and deposition), transformation and fate of mercury and
mercury compounds" (UNEP, 2013). Given the atmosphere represents the primary pathway for
the global distribution of mercury (Schroeder et al. 1998; Selin 2009; Driscoll et al., 2013), highly
accurate and precise atmospheric monitoring of Hg is paramount in attaining these goals.
Existing atmospheric Hg monitoring networks, such as the Global Mercury Observation System
(GMOS), the Atmospheric Monitoring Network (AMNet), and the Environment and Climate
Change Canada-Atmospheric Mercury Monitoring (ECCC-AMM) network, have greatly improved
understanding of atmospheric Hg (Li and Lee, 2014) and the ability to develop and evaluate
global atmospheric distribution models (Lin et al., 2006) such as GEOS-Chem, GLEMOS, GEM-
MACH-Hg, and ECHMERITRADM (Travnikov et al., 2017). However, the spatial scope of these
networks is limited, especially in the southern hemisphere (Li and Lee, 2014), leading to
considerable gaps in the understanding of atmospheric Hg cycling. It is widely acknowledged that
further network expansion will be required (Pirrone et al., 2013; Sprovieri et al., 2017).

The principal constraints limiting the spatial expansion of atmospheric Hg measurements are
high costs and dependence on electricity, compressed gases and technical training (McLagan et
al., 2016a; Pirrone et al., 2013; Huang et al., 2013a). Passive air samplers operate without these
constraints and have the potential to complement existing approaches and greatly improve the
spatial resolution of measurements (McLagan et al., 2016a; Pirrone et al., 2013; Huang et al.,
2013a). One successful example of such a combined approach is the Global Atmospheric Passive
Sampling (GAPS) Network for persistent organic pollutant (POP) monitoring. Shortly after the
implementation of the Stockholm Convention on POPs, the GAPS network was established to
"complement high-volume, active air sampling activities in assessing the presence of POPs in the
atmosphere and in evaluating their global distribution and long-range transport" (Pozo et al.,
2006). The network now includes over 40 sites across all seven continents (Pozo et al., 2006;
Shunthirasingham et al., 2010; Herkert et al. 2018).



McLagan et al. (2016b) calibrated a passive air sampler for measuring gaseous Hg in the atmosphere that utilizes a Radiello® diffusive barrier and a sulphur-impregnated activated carbon sorbent. The sampler's replicate precision (2%) is excellent (McLagan et al., 2016b). Also

the variability of its sampling rate (*SR*; volume of air effectively stripped of gaseous Hg per unit time) caused by meteorological parameters is small, increasing slightly with both temperature and wind speed across ranges relevant to outdoor deployments (McLagan et al., 2017b). However, testing thus far has been restricted to deployments in laboratory settings and at only one outdoor location. To better understand the PAS's overall uncertainty, its performance under

variable geographical, meteorological, and gaseous Hg concentrations must be evaluated.

In this study, we seek to assess the accuracy of the McLagan et al. (2016b) PAS by comparing ambient Hg concentrations derived from the PAS to those measured with established active sampling techniques. PASs were deployed at sites with on-going active monitoring instruments in North America, Asia, Australia, and Europe. These sites cover a wide range of meteorological

conditions and some variation in gaseous Hg concentrations. In addition to quantifying the overall uncertainty of the PAS, this study also allowed for the refinement of the previously calibrated *SR* using a much larger pool of data. Furthermore, some of the selected sites recorded the speciation of atmospheric Hg, making it possible to investigate whether gaseous oxidized Hg (GOM) is being taken up by the PAS.

## 2. Methods

### 2.1 Passive air sampling

The PAS used in this study has been described in detail by McLagan et al. (2016b). Briefly, a stainless-steel mesh cylinder filled with sulphur-impregnated activated carbon (HGR-AC; Calgon Carbon Corporation) is placed inside a white Radiello® diffusive barrier (Sigma Aldrich). This

barrier is protected from wind and precipitation by attachment to the inside of a protective shield, which also serves as a storage and shipping container. The sampler works by diffusive uptake and accumulation of gaseous Hg onto the sorbent. Following deployment, the sampler is retrieved, the sorbent contents are analyzed on an automated thermal combustion atomic absorbance instrument (McLagan et al. 2017) and the time-averaged gaseous Hg concentration is

calculated using a previously calibrated sampling rate (see Section 2.5).



The term gaseous Hg is used to describe the sorbed analyte because it has not been confirmed whether this PAS takes up gaseous elemental Hg (GEM) or TGM (total gaseous Hg; both GEM and GOM). It is, however, unlikely that the highly reactive nature of GOM allows it to pass through the pores of the diffusive barrier. The most recent modelling estimations suggest that the "effective

lifetime" of GEM is around 6 months (Corbitt et al., 2011; Horowitz et al., 2017). The atmospheric lifetime of GOM due to reduction and deposition is on the order of days to weeks (Ariya et al., 2015; Horowitz et al., 2017; Shah et al., 2016). Although uncertainties remain, the shorter atmospheric lifetime and higher deposition fluxes of GOM translate to GEM making up the majority of TGM in most places (typically >95%) (Cole et al., 2014; Driscoll et al., 2013; Rutter et

al., 2009; Slemr et al., 2015). As such any uncertainty related to the uptake of GOM by the PASs is likely small.

For this study, PASs and standard operating procedures (see Section S1) were sent from Toronto to 20 sampling sites on four continents (Fig. 1) using Canada Post or international couriers. The PAS were deployed for a period of one year at each site during the time between 2015 and 2017.

Accessibility of AMNet and ECCC-AMM networks resulted in a greater number of sites in North America than in other global regions. Temporal resolution of sampling ranged from monthly, quarterly, bi-annual to annual deployments. The number of deployments varied between sites. Four sites with a deployment intensity categorized as "high" were sampled with monthly, quarterly, bi-annual, and annual resolution. Six "moderate" deployment intensity sites had

quarterly, bi-annual and annual resolution. Seven "low" deployment intensity sites had bi-annual and annual resolution and three "very low" deployment intensity sites had year-long deployments. Exact numbers, lengths and dates of deployments at each site are shown in Table S2.1, Table S2.2, and Table S2.3. After deployments, PAS were stored at each site until the last PAS had been retrieved, at which point they were returned to Toronto by courier for analysis of

the sorbent content. In total, there were 142 triplicated deployments (426 samplers) across all sites. Field blanks were obtained by transporting PASs to each site, removing the Teflon tape and solid cap, attaching an open cap with mesh screen, holding it up to a deployment position for 10 s, and then immediately taking it down, closing and sealing it as described above. Temperature and wind speed for each deployment period, as recorded by weather stations at or near the

sampling sites, are listed in Table S2.3.

## 2.2 Active air sampling


TGM/GEM concentrations were measured with Tekran 2537 series cold-vapour atomic fluorescence spectrometer (CVAFS) systems. Details of instrument setup are given in Landis et al. (2002), Steffen et al. (2008), Cole et al. (2013), and AMNet (2015). Concisely, the instrument
consists of a 2 μm Teflon filter (47 mm diameter) at the inlet of a heated sampling line (of variable length; site dependent) and a second filter at the instrument inlet. Australian sites did not use the outer filter or a heated inlet and are therefore likely to only sample GEM. Ambient air is pulled through the sampling line at a flow rate of 1 to 1.5 L min$^{-1}$ (depending on site and network protocols) with concentrations normalized to the specific rate at each site. Continuous,
5-minute resolution concentrations are produced by alternating collection and analysis on one of the two gold traps within the instrument. The analysis phase is initiated by heating a gold trap to ≈ 500 °C, releasing the sorbed Hg into pure argon gas, and detecting mercury by atomic florescence spectrometry. Routine internal (permeation source) and external (injections) calibrations were performed at each site. All data were quality controlled using the Research
Data Management and Quality Assurance System (RDMQ™; McMillan et al., 2000; Steffen et al., 2012). Differences in sampling methods may result in slight inconsistences in the sampled species among sites.

Tekran 1130/1135 speciation units paired with a Tekran 2537 CVAFS were deployed at Alert, Mauna Loa, Salt Lake City, Beltsville, and Grand Bay. Again, full details are provided elsewhere
(Landis et al., 2002, Steffen et al., 2008, Cole et al., 2013). In brief, ambient air enters these systems through an impactor inlet (to remove particles > 2.5 μm) before passing through a potassium chloride denuder to trap GOM, a particle filter to collect particulate bound Hg, and finally into the Tekran 2537 for analysis of GEM. The typically low concentrations of GOM and particulate bound Hg require a higher flow rate (10 L min$^{-1}$) and a lower temporal resolution (1
to 3-hours) to collect sufficient analyte for analysis by the Tekran 2537. The GEM component is continuously measured at 5-minute resolution in the same manner described for the Tekran 2537 series samplers above. It has been hypothesized that GOM may make a greater proportion of TGM than previously thought (Ambrose et al., 2013; Gustin et al., 2015; Huang et al., 2013b). Nonetheless, the incipient GOM measurement techniques (based on ion exchange membranes or
nylon filters) used have produced inconclusive results and, similar to the conventional sampling methodologies (based on particle impact filters and annular denuders), they are subject to sampling artefacts that may skew results (Cheng and Zhang, 2016; Ariya et al., 2015). Thus, the





GOM measurements made at these sites using Tekran speciation analyzers may still aid in elucidating whether the samplers sorb solely GEM or TGM.

Information on the percentage data coverage across the deployment periods of the active measurement systems and PASs (successful samples/deployments out of actual samples/deployments) is shown in Table S2.2. The 5-min TGM/GEM data from the Tekran 2537 series instruments or the Tekran 2537 component of the speciation units were compared with the concentrations derived from the PASs. For Alert station, which operates both a Tekran 2537X

sampler and a Tekran 1130/1135 speciation unit, data from the former were used for comparison.

Given that TGM is generally dominated by GEM at most sites (typically >95%) (Cole et al., 2014; Driscoll et al., 2013; Rutter et al., 2009; Slemr et al., 2015), differences between the two concentrations are likely to be small. To ensure consistency with the nomenclature used for the

PASs, the analyte sampled by non-speciating Tekran 2537 instruments is referred to as gaseous Hg. The PAS derived concentrations were compared with the speciated data available for Alert, Mauna Loa, Salt Lake City, Beltsville, and Grand Bay to ascertain whether the PASs accumulate GOM.

Mauna Loa (3397 m a.s.l.) and Mt. Lulin (2862 m a.s.l.) are high altitude sites and required active

concentrations to be adjusted for pressure. At Mt. Lulin, to obtain a volumetric flow rate of 10 L·min$^{-1}$, the mass flow rate was lowered by a factor of 0.6900, which is the ratio of the average atmospheric pressures at this site (70.00 kPa) and at sea level (101.325 kPa). The gaseous Hg concentrations reported for Mauna Loa were multiplied by 0.6716, which is the ratio of the mean air pressure during the deployments (68.05 ± 0.04 kPa) and at sea level.

**2.3 Data reporting**

The uncertainty of all reported values is given by one standard deviation. The standard errors of regression coefficients were also converted to standard deviations. Uncertainties are reported to one significant digit unless the first non-zero digit is a 1, in which case there are two significant digits (e.g. 5.43 ± 0.17; Hughes and Hase, 2010). All data are written with one extra digit beyond

the appropriate significant digits to avoid rounding errors when using the PAS in the future. The extra digit is written in subscript (i.e. $0.345_0 ± 0.003_4$ or $5.42_9 ± 0.17$). The exception to the use of



the added digit is the reporting of gaseous Hg concentrations, which will follow standard reporting methods used in the atmospheric Hg literature: measurements corrected to the nearest 0.01 ng m$^{-3}$.

**2.4 Analysis for total Hg**

All PASs were analyzed at the University of Toronto Scarborough for the total mass of Hg using an AMA254 (Leco Instruments Ltd.) by means of thermal combustion, amalgamation, atomic absorption spectroscopy in pure oxygen carrier gas (USEPA Method 7473; USEPA, 2007). To prevent sulphur poisoning of the instrument's catalyst by the high sulphur sorbent (S = 8-15

wt%), sodium carbonate was added to the end of the catalyst tube ($\approx$ 5 g) and directly on top all analyzed samples, standards, and reference materials ($\approx$ 0.2 g) (McLagan et al., 2017a). The entire mass of sorbent in each sample (0.73 ± 0.04 g) was analyzed in two aliquots of up to 0.45 g to remove uncertainty related to the heterogeneity of sorbed Hg throughout the sorbent matrix. During analysis, samples were dried at 200 °C for 30 seconds then combusted at 750 °C for 330

seconds. After reduction by the system catalyst, which was continually heated to 550 °C, GEM was trapped on a gold amalgamator. After combustion, a 60 second purge ensured the removal of all pyrolysis gases from the system. Heating the amalgamator to 900 °C for 12 seconds released the trapped Hg into a cuvette where absorption was measured by dual detector cells for high and low absolute amounts of Hg at 253.65 nm wavelength.

The instrument was calibrated via the addition of diluted Hg liquid standard (1000 ± 5 mg L$^{-1}$; in 10% w/w HCl; Inorganic Ventures) to $\approx$ 0.22 g of clean (unexposed) HGR-AC sorbent. The low and high cells were calibrated using standards of 0, 1, 2.5, 5, 10, 15, 20 ng and 25, 50, 100, 250, 500 ng, respectively (uncertainty in autopipette is 1 ± 0.004 ng). As the response is not linear near each cell's upper limit ($\approx$ 20 – 25 ng for the low cell and 500 ng for the high cell), calibration

curves were fitted with quadratic relationships.

2.4.1 Quality assurance/quality control.

Analytical blanks, i.e. clean (unexposed) HGR-AC sorbent, were analyzed regularly and had a mean concentration of 0.30$_9$ ± 0.14$_7$ ng g$^{-1}$ ($n$ = 33). The mean field blank concentrations for each sampling site are listed in Table S2.4. All PAS deployments were blank adjusted by subtracting

the mean field blank Hg concentration (ng g$^{-1}$) at each site multiplied by the mass of HGR-AC in





the sample from the mass of Hg in the sample. Analytical precision was examined by analyzing 5, 10, 50, and 100 ng Hg liquid standards added to ≈ 0.22 g of clean (unexposed) HGR-AC approximately every 10 – 15 samples. Recoveries of the Hg liquid standards were $99.9_5 \pm 1.1$ % ($n$ = 215). Accuracy was monitored via alternating analysis of a high sulphur, bituminous coal

standard reference material, NIST 2685c (S = 5 wt%; National Institute of Standards and Technology) and our own in-house reference material RM-HGR-AC1 (powdered HGR-AC sorbent loaded with Hg by exposure to air for four months then homogenized; $23.1 \pm 0.8$ ng g$^{-1}$ based on 198 analytical runs) approximately every 10 – 15 samples. Recoveries for the NIST 2685c and RM-HGR-AC1 were $98._4 \pm 2._7$ % ($n$ = 57) and $98._8 \pm 3._6$ % ($n$ = 86), respectively.

**2.5 Calculation of air concentration from the amount taken up by the PAS**

Gaseous Hg concentrations in the atmosphere, $C$ (ng m$^{-3}$), are calculated from the mass of sorbed Hg, $m$ (ng), according to Eq. 1:

$$C = {m}/{(SR \cdot t)} \tag{1}$$

where $SR$ is the sampling rate of the PAS (McLagan et al., 2016b) and $t$ is the deployment time of

the sampler (day). In this study, three sets of air concentrations were derived from the measured $m$. The first set was derived by using the original, published $SR$ of $0.121_0$ m$^3$ day$^{-1}$ (hence termed: *original SR*) obtained during a year-long calibration of the PAS on the campus of the University of Toronto Scarborough (McLagan et al., 2016b).

The second set of air concentrations was calculated by using a recalibrated $SR$ obtained from the

data generated in this study plus the data from the original calibration experiment (hence termed: *recalibrated SR*). This *recalibrated SR* was calculated using the slope method as described by Restrepo et al. (2015) and McLagan et al. (2016b). Rearranging Eq. 1, we can derive a $SR$ for individual deployments from the mass of sorbed Hg, $m$, in a PAS and the gaseous Hg concentration, $C_A$, measured by an active instrument during that PAS's deployment period:

$$SR = {m}/{(C_A \cdot t)} \tag{2}$$

In this method, $SR$ is calculated as the slope of a linear regression of $m$ against ($C \cdot t$).





The third set of air concentrations was calculated using the *recalibrated SR* with adjustments for the mean temperature, $T_{exp}$ (°C), and wind speed, $WS_{exp}$ (m s⁻¹), (hence termed: *adjusted SR*) during each deployment using factors previously determined in controlled laboratory

experiments (Eq. 3; McLagan et al., 2017b):

$$SR_{adj} = SR_{cal} + \left(T_{exp} - 9.89\,°C\right) \cdot 0.0009\, \frac{m^3}{day\,°C} + \left(W_{exp} - 3.41\, \frac{m}{s}\right) \cdot 0.003 \frac{m^2\,s}{day} \qquad (3)$$

where $SR_{cal}$ is the recalibrated *SR* (m³ day⁻¹).

The three sets of air concentrations derived from the PAS, $C_{PAS}$ (ng m⁻³), were compared with the actively derived gaseous Hg concentrations, $C_{ACT}$ (ng m⁻³), and in each case a mean normalized

difference (MND) was calculated:

$$Mean\ normalized\ difference\ (MND)\ \% = \left(\frac{1}{n}\sum_{i=1,n}\frac{|C_{ACT,i} - C_{PAS,i}|}{C_{ACT,i}}\right) \cdot 100 \qquad (4)$$

where *n* is the number of comparisons. Deployments were included in the MND calculation if at least one PAS was successfully deployed and analyzed and successful active sampling data covered at least 25 % of the PAS's deployment period. To determine if passive concentrations

were improved by the temperature and wind speed *adjusted SR* the MND of the *recalibrated SR* and *adjusted SR* were compared by means (one-tailed pairwise T-test) and variance (Levene's Test). For sites with active speciation measurements, MNDs based on active – passive comparisons of GEM and TGM data were compared for both the *recalibrated SR* and *adjusted SR* at each site (two-tailed pairwise T-test) to determine which data set is a better fit with the

passive concentrations. The mean relative standard deviation (RSD; standard deviation divided by the mean multiplied by 100) of replicates from individual deployments was used to assess the precision-based uncertainty of the sampler.

## 3. Results and Discussion

### 3.1 Replicate precision of the passive air sampler

Few PAS samples were lost during deployment, transport or analysis. Reasons for losses were poorly sealed samplers, errors in recording of deployment time and dates, loss during analysis (e.g. catalyst failure), and a hail storm (Hunter Valley site). Of 142 triplicated PAS deployments,



93 % (132 deployments, 378 samplers) and 89 % (129 deployments, 375 samplers) had at least one and two successfully analyzed PAS, respectively. The precision-based uncertainty of the PAS

calculated from the successful replications was $3._6 \pm 3._0$ % (Table 1), which is slightly worse than was reported for the original outdoor calibration experiment (2 ± 1 %) in Toronto, Canada (McLagan et al., 2016b). The slight decrease in precision is to be expected as the samplers were deployed by different individuals at each location whose only training were written and video-recorded standard operating procedures (see Section S1). Nonetheless, the precision of the

sampler remains high, especially in comparison to other PAS for Hg, the best of which had a reported precision of 7.7 % (Skov et al., 2007). Others had deviations between replicates in excess of 10 % or precision was not reported at all (Brown et al., 2012; Gustin et al., 2011; Huang et al., 2012; Zhang et al., 2012). These results confirm the excellent reproducibility that can be achieved with this PAS even when used by newly and informally trained personnel.

**3.2 Passive air sampler uptake curves**

The amount of mercury quantified in each individual sampler is plotted against the deployment time in Fig. 2. These uptake curves are highly linear over 12-months at all sites, confirming that the PASs do not approach a limit to their uptake capacity throughout all deployments. At Xiamen and Ningbo, sites with the highest uptake rates, i.e. the mass of Hg sorbed per unit time, and

hence the highest ambient gaseous Hg concentrations (Table 1), 144 and 133 ng of Hg, respectively, was taken up over a 12-months period. This is almost double the mass of Hg taken up in the original uptake experiment (McLagan et al., 2016b) and indicates the maximum deployment time of the sampler is at least one year even under the elevated concentrations observed in East Asia. The high uptake capacity of the HGR-AC sorbent for gaseous Hg is due to a

large surface area to volume ratio and the affinity of gaseous Hg to the impregnated sulphur (McLagan et al., 2016a; Suresh Kumar Reddy et al., 2013; Zhang et al., 2012).

Uptake rates can be derived from the slopes of the curves in Fig. 2. Uptake rates of PASs deployed at the same site (comparing slopes within each panel in Fig. 2) are very uniform. The greatest relative variability in uptake rate between samples at any one site occurred at Alert (Table 1) and

was low (<25 % RSD; Table 1). This attests to the stability of both (i) the *SR* and (ii) the gaseous Hg concentrations at each site over the length of the PAS deployments (1 month or longer). The time-averaged nature of the concentrations measured by the PASs conceals much of the



variability that generally occurs at shorter time resolution. The higher variability in uptake rates and gaseous Hg concentrations at Alert and Ningbo (Table 1) can be attributed to spring-time atmospheric Hg depletion events (Steffen et al., 2008) and seasonal variability in the elevated East Asian background concentrations, respectively. The differences in uptake rates between sites (comparing slopes among different panels in Fig. 2) were caused by different gaseous Hg concentrations at each location as evidenced by the significant correlation between uptake rate and active gaseous Hg concentration data ($p < 0.001$).

## 3.3 Sampling rates and differences between active- and passive-derived gaseous Hg concentrations

The Tekran 2537 active monitoring instruments successfully recorded Hg concentrations during 59 % of all deployment periods. Data gaps were caused by instrument failures, power outages, or removal of poor quality data during RDMQ™ processing. The PASs covered a greater percentage of the deployment periods across all sites, indicating high reliability and ease-of-use. A comparison of active and passive sampling was considered meaningful if an active instrument had recorded data during at least a quarter of a PAS's deployment time. This was the case in 113 of 142 deployments (80 %). 107 of these 113 deployments (306 samplers) had at least one successful PAS data point, and therefore were used in the comparison of actively and passively derived gaseous Hg concentrations (Table 1). Because the sites in Sydney, Hunter Valley, and Xiamen, did not have adequate data coverage for any of the deployments, the comparison included 17 of the 20 total sites.

The gaseous Hg concentrations calculated from the PAS using the *original SR* of $0.121_0 \pm 0.005$ m$^3$ day$^{-1}$ (McLagan et al., 2016b) deviated from the active air concentrations on average by $14._2 \pm 10$ %. The PASs were, in general, overestimating the gaseous Hg concentrations (most data points and the linear regression line between active and passively determined concentration are above the 1:1 line in Fig. 3A), which suggested the *original SR* was biased low. Laboratory experiments on the effects of meteorological parameters on the *SR* also suggested the *original SR* was low (McLagan et al., 2017b). The *original SR* was based on only 37 samples at one sampling location in Toronto (McLagan et al., 2016b). The limited data set and range of conditions is the likely reason for much of this bias.




The *recalibrated SR* based on 343 samples deployed at 17 sites with collocated active and passive sampling data (37 samples from McLagan et al., 2016a and 306 samples from this study) was $0.135_4 \pm 0.016$ m$^3$ day$^{-1}$ (see Fig. S2.1 for plot of $m$ against $C \cdot t$ for all samples). Because of the

broad range of latitudes, meteorological conditions, deployment times, and altitudes across these sites, we recommend using the *recalibrated SR* of $0.135_4 \pm 0.016$ m$^3$ day$^{-1}$ as the baseline *SR* for the PAS.

McLagan et al. (2016b) theoretically estimated the *SR* of the PAS based on the molecular diffusivity of elemental Hg and an estimated effective diffusion distance. Using an air-side

boundary layer thickness of 15 mm, which has been recommended for outdoor deployments with the protective shield (McLagan et al., 2016b) and the mean temperature across all deployments in this study ($9.8_9$ °C), this approach predicts a *SR* of $0.130_6$ m$^3$ day$^{-1}$. This modeled *SR* is only 3.5 % lower than the *recalibrated SR*, which provides further confidence in the mechanistic understanding of the uptake process of Hg within this PAS.

When the *recalibrated SR* is used for the derivation of air concentrations from the PAS, the MND compared to active instrument derived concentrations is significantly ($p < 0.001$) reduced to $8.8 \pm 7.3$ %. When the air concentrations for the PAS are derived using the *adjusted SR*, the MND is $8.7 \pm 5.7$ % (Table 1). While this value is not significantly lower than that obtained using the *recalibrated SR* ($p = 0.581$), passive and active gaseous Hg concentrations are more highly

correlated when the adjusted *SR*s were applied (R$^2$ reported in Fig. 3). Moreover, the variance in the discrepancy is significantly lower (Levene's Test; $p = 0.046$), which can be linked to smaller discrepancies when the *adjusted SR* was applied at some of the sites with extreme wind speed and/or temperature conditions. For example, without adjustment the gaseous Hg concentrations at the Cape Grim site were overestimated by the PAS as the *recalibrated SR* (unadjusted) is too

low for the strong westerly wind at this site (mean: 10.2 m s$^{-1}$ substantially higher than at any other site, Table S2.3). Not surprisingly, at locations with conditions closer to the mean temperature ($9.8_9$ °C) and wind speed ($3.4_1$ m s$^{-1}$) of the adjusted *SR* calculation, differences between active and passive sampler derived concentrations were very similar when the *recalibrated SR* or the *adjusted SR* was applied (Table 1). We also calculated gaseous Hg

concentrations for the PAS using the *recalibrated SR* adjusted solely for temperature and solely for wind speed. In neither case did these individually adjusted *SRs* significantly reduce the discrepancies or their variance ($p > 0.05$) relative to the use of the unadjusted *recalibrated SR*.





### 3.4 Placing the PAS performance into context

It is important to acknowledge that this is not a fully independent evaluation of the performance
of the sampler, as the same data were used in the derivation of the *recalibrated SR* and in the
determination of the air concentrations from the PAS. However, we stress that data from all sites
were used to derive a single *recalibrated SR* that was used for all sites, i.e. the fitting involved was
not site-specific. For example, the Little Fox Lake site contributed the most data points to the
recalibration ($n$ = 49), but those data only represented 14 % of the whole data set. Thus, concerns
over this issue are minimal.

When assessing the accuracy or overall uncertainty of the PAS, we must also consider the
inherent uncertainty of active monitoring instruments. By sampling with two collocated Tekran
2537A instruments, Temme et al. (2007) estimated a measurement uncertainty of 8.8 %. Aspmo
et al. (2005) describe the uncertainty of the same system in the range of 5-10 %. Based on a
review of several inter-comparisons involving Tekran 2537 instruments, Slemr et al. (2015)
estimated a systematic uncertainty of ≈10 %, but warn this can expand up to 20 % in extreme
cases. Calibrating the PASs using the Tekran data incorporates the active instrument uncertainty
into the passive data and does therefore account for some of the PAS inaccuracies. Additionally,
at all sites, there were data gaps in the active instrument measurements (see Table S2.2) that if
successfully analyzed would have altered the active concentration at that site in some way; minor
or otherwise. As such, we can conclude that the MND across all sites using the *adjusted SR* ($8.7 \pm$
$5.7$ %) represents the maximum uncertainty of measurement with the PAS. Even this
conservative assessment of PAS accuracy is in line with active instruments uncertainty and
qualifies the device as appropriate for background monitoring of gaseous Hg. The excellent PAS
precision again highlights the sampler consistency; $3.6 \pm 3.0$ % is lower than the replicate
precision of collocated active instruments.

The performance of the PAS using either the *recalibrated* or *adjusted SR* represents a substantial
improvement over all existing gaseous Hg PAS designs to date, especially those with sufficiently
low detection limits to monitor background concentrations (as summarized in a review on
gaseous Hg PASs by McLagan et al., 2016a). While the accuracy-based uncertainty of the 3M PAS
by McCammon and Woodfin (McCammon and Woodfin, 1977) was similar ($8 \pm 7$ %), this device
was only tested in the range of 25,000 – 300,000 ng m$^{-3}$ gaseous Hg concentrations over an 8-



hour period, making it unsuitable for background monitoring. Of the PASs that have sufficiently low detection limits to monitor background concentrations the lowest overall uncertainties were

19 ± 14 (Huang et al., 2012) and 22 ± 15 % (Guo et al., 2014; Zhang et al., 2012). Other designs had uncertainties greater than 30 % (Brown et al., 2012; Nishikawa et al., 1999), reported only replicate precision (Brumbaugh et al., 2000; Skov et al., 2007), or reported no uncertainty estimate at all (Gustin et al., 2011).

**3.5 Site specific analysis**

Plots comparing active instrument derived gaseous Hg concentrations with passive concentrations determined using the *original*, *recalibrated*, and *adjusted SRs* for each sampling site are presented in Section S3 (Fig. S3.1 – S3.17).

3.5.1 Urban Sites

Of the 20 sampling locations from the current study, five were classified as urban (Xiamen,

Ningbo, Salt Lake City, New York City, and Sydney). Additionally, the previous calibration study site in Toronto was included in recalibrations and uncertainty assessments. Overall, there was good agreement between active and passive concentrations using the *recalibrated* (MND: $8.5 \pm 4.7$ %) and *adjusted SRs* (MND: $8.5 \pm 5.2$ %) for those five sites. Both Xiamen and Ningbo sites were selected due to the elevated gaseous Hg concentrations typically observed in East Asia (Wan et

al., 2009; Xu et al., 2015; Zhu et al., 2012). Unfortunately, the majority of active sampling data at Xiamen and nearly half the data at Ningbo were lost due to active instrument malfunctions, limiting active–passive comparisons to 10 deployments at Ningbo. The mean temperature and wind speed during deployments at this site (Table S2.3) varied little from the mean values across all sites, which resulted in little difference between passive concentrations derived from

*recalibrated* and *adjusted SRs* (Table 1; Fig. S3.7). The active data recorded in Sydney, Australia were also insufficient across any of the deployments for comparison with passive concentrations. Mean gaseous Hg concentrations calculated using the *adjusted SR* for all PAS deployments were $2.53 \pm 0.28$ in Xiamen and $0.91 \pm 0.28$ ng m$^{-3}$ in Sydney.

Salt Lake City may experience elevated GOM concentrations and atmospheric Hg reactivity in

general due to the increased presence of atmospheric halogenated species in the atmosphere around the Great Salt Lake (Gay et al., 2013; Peterson and Gustin, 2008; Stutz et al., 2002). Only


one of the seven deployments in Salt Lake City had sufficient actively measured speciation data for comparison. The mean GOM concentration for that deployment was 0.014 ng m$^{-3}$, which represents <1 % of the mean active GEM concentration for the same period (1.67 ng m$^{-3}$). Hence,

we are unable to use this data to infer the mercury species sampled by the PAS. Discrepancies between active instrument derived concentrations and concentrations derived using *recalibrated* and *adjusted SR*s over this single deployment were not significantly different ($p$ = 0.280).

Concentrations derived from PASs deployed in New York City using *recalibrated* (8.$_6$ ± 4.$_6$) and *adjusted SRs* (7.$_7$ ± 4.$_6$) deviated to a similarly small extent from the active concentrations ($p$ =

0.271; Table 1; Fig. S3.4). While mean temperatures during deployment were higher than the overall mean, wind speeds were lower, cancelling out their respective effects (Table S2.3). Differences between active and passive derived concentrations at Toronto from the previous study were also similar using both *recalibrated* and *adjusted SRs* (10.$_2$ ± 4.$_4$ and 10.$_6$ ± 5.$_2$ %, respectively). A full discussion of those results can be found in McLagan et al. (2016b).

Good agreement between active and passive concentrations both at typical hemispheric and elevated East Asian background concentrations using either the *recalibrated* or *adjusted SR* (uncertainties not significantly different; $p$ = 0.381) demonstrates that the *SRs* are not concentration dependent, which is essential for effective gaseous Hg monitoring.

### 3.5.2 Rural Sites

11 sites (Beltsville, Put-In-Bay, Grand Bay, Kejimkujik, Ucluelet, St. Anicet, Egbert, Waldhof, Hunter Valley, Cape Grim, and Gunn Point) were located in rural settings. Put-In-Bay, is situated on the shores of Lake Erie in Ohio, USA. Temperatures at the site were similar to the mean of all sites, but wind speeds were high. Nonetheless, differences from active concentrations were small using either *recalibrated* or *adjusted* SR, i.e., were not significantly reduced by *adjusting* the *SR* for

wind speed and temperature ($p$ = 0.082). Grand Bay (on the Gulf of Mexico in Mississippi, USA) experiences relatively high temperatures and moderate wind speed (Table S2.3). Using the *adjusted SR* significantly reduced the discrepancy ($p$ = 0.013) from the active data compared to the use of the *recalibrated SR* (Fig. S3.2). Two sites are situated in the very North and South of Australia; Gunn Point is a hot, windy location at the top of the Northern Territory and Cape Grim

is located on the mild, but exceptionally windy west coast of Tasmania (Table S2.3). Consequently, these more extreme conditions resulted in the calculation of the highest mean





*adjusted SR* values of any of the sites ($0.157_2 \pm 0.000_6$ and $0.157_2 \pm 0.002_0$ m$^3$ day$^{-1}$, respectively).
The concentrations derived from the *adjusted SR* were substantially reduced relative to those
calculated using the *recalibrated SR* (Fig. S3.9 and S3.10, respectively), which resulted in a

significantly better agreement with the active concentrations at both Cape Grim ($p < 0.001$) and
Gunn Point ($p = 0.019$). Significant improvements in passive concentration data at three of the
four sites with high temperatures and/or wind speeds demonstrate the need for adjusting the *SR*
during similar deployments. While there was some seasonal variability in conditions at Beltsville,
Kejimkujik, St. Anicet, Ucluelet and Waldhof, mean temperature and wind speeds at each site

were, in general, similar to the mean values for all sites. As such, there were no significant
differences between MNDs with either the *recalibrated* or *adjusted SRs* ($p > 0.05$; Table 1; Fig.
S3.1, S3.13, S3.15, S3.14, and S3.6, respectively) at these locations.

At both Beltsville and Grand Bay, the proportion of GOM in TGM measurements was <1 % for all
deployments. Hence, no information on the sampled analyte could be derived from these data.

3.5.3 High Altitude Sites

The two high altitude sampling sites at Mt. Lulin and Mauna Loa provided a unique opportunity
to not only examine the PAS's functionality under relatively low atmospheric pressure
conditions, but also to test its performance in a likely more dynamic zone of atmospheric
chemistry at or above the planetary boundary layer (Bieser et al., 2017; Carbone et al., 2016).

Lower atmospheric pressure has the potential to affect the PAS in two ways: (i) increasing the *SR*
because diffusivity coefficients are inversely proportional to pressure (Armitage et al., 2013;
Klánová et al., 2008; Seethapathy et al., 2008) and (ii) decreasing the *SR* as there is less mass per
volume of air. While the use of mixing ratios would prevent the latter effects, they are not the
preferred method of reporting atmospheric Hg measurements (Weigelt et al., 2016). These two

effects should theoretically cancel each other out, hence we would expect passive concentrations
to align with pressure-adjusted active concentrations. At Mt. Lulin the MNDs were low for both
the *recalibrated* ($5.3 \pm 1.2$ %) and *adjusted* ($6.7 \pm 1.2$ %) *SRs* and not significantly different ($p = 0.118$; Fig. S3.8). There was also no significant improvement in MND ($p = 0.693$) using the
*adjusted SR* ($12._2 \pm 4._6$ %) over the *recalibrated SR* ($11._7 \pm 4._5$ %) at Mauna Loa (Fig. S3.17). While

the MNDs were higher at Mauna Loa than at Mt. Lulin, these values are relative to the observed
gaseous Hg concentrations, which were low at Mauna Loa (Table 1). In absolute terms, the mean



differences between active and *recalibrated* and *adjusted* passive concentrations at Mauna Loa were 0.11 ± 0.04 and 0.10 ± 0.04 ng m$^{-3}$, respectively.

Active GOM concentrations measured by the Tekran speciation unit were elevated at Mauna Loa and made up $9._5 \pm 2._9$ % of TGM concentrations measured by the same system. Adding GOM concentrations to the 5-minute resolution GEM measurements used in the active – passive comparisons significantly increased MNDs for both the *recalibrated SR* ($p < 0.001$, TGM: $20._4 \pm 2._9$ %; GEM: $11._9 \pm 4._5$ %) and *adjusted SR* ($p < 0.001$, TGM: $20._7 \pm 2._8$ %; GEM: $12._2 \pm 2._6$ %). These data, considered in isolation, suggest the PAS is not taking up GOM.

### 3.5.4 Northern/Arctic Sites

Two sites from Canada's north were included in the study: Alert (Fig. S3.16), within the Arctic Circle, and Little Fox Lake (Fig. S3.11), which is north of Whitehorse in the Yukon Territory. Both sites had high temporal resolution data, except for the first four months of sampling (October to February) at Alert, when PAS data were lost due to poorly sealed samplers and contaminated

field blanks. While wind speeds were moderate and not excessively variable across deployments at either site, the mean temperatures of each deployment ranged over 27.4 K at Little Fox Lake and 20.5 K at Alert (Table S2.3). Despite the larger range of temperatures at these sites, mean temperatures across all deployments were not excessively low (Alert: 5.9 °C; Little Fox Lake: 2.2 °C) as northern summer temperatures were relatively high at both sites. Little Fox Lake, the site

with the greatest temperature range, had a significant improvement in the MND using the *adjusted SR* ($p = 0.027$), whereas Alert did not ($p = 0.454$). Although the reduction in MND between active and passive derived concentrations at the Northern/Arctic site with the greatest temperature range (Little Fox Lake) highlights the benefit of *adjusting SR* under extreme conditions, at both sites the MND for either the *recalibrated* or *adjusted SR* was low (< 7 %).

The Alert site also employed a Tekran speciation system. The mean GOM concentrations at Alert across the different PAS deployments represented $4._7 \pm 6._5$ % of the TGM measured by the speciation unit. The high variability in the GOM proportion is associated with spring-time atmospheric Hg depletion events; the mean proportion of GOM in TGM was $13._7 \pm 13$ % during the Spring (March, April, and May) deployments. When we consider the data from just these

deployments ($n = 6$) there was no significant difference between MNDs based on the *recalibrated SR* ($p = 0.280$; TGM: $14._0 \pm 9._3$ %; GEM:$21._1 \pm 12$ %) or the *adjusted SR* ($p = 0.140$; TGM: $13._7 \pm 8._8$



%; GEM: $22._8 \pm 11$ %). When all the data from Alert was considered ($n = 36$), there again was no significant difference between MNDs based on the *recalibrated SR* ($p = 0.097$; TGM: $6._8 \pm 6._7$ %; GEM: $9._9 \pm 8._8$ %) or the *adjusted SR* ($p = 0.065$; TGM: $7._0 \pm 6._0$ %; GEM: $10._2 \pm 8._4$ %). Taken on its

own this data cannot confirm whether the PASs are taking up solely GEM or both GEM and GOM (TGM).

## 4. Recommendations and Conclusions

From this much larger data set of collocated active and passive measurements of gaseous Hg we were able to revise our *original SR*, which we determined to be overestimating concentrations, to

a *recalibrated SR* of $0.135_4$ m$^{-3}$ day$^{-1}$. The variability of the maximum uncertainty of the PAS ($8._7 \pm 5._7$ %) was improved by the application of a temperature and wind speed *adjusted SR* (Eq. 3) recommended by McLagan et al. (2017b). This is in the same range as uncertainties attributed to active measurement instruments (Aspmo et al., 2005; Slemr et al., 2015; Temme et al., 2007) and is unprecedented in gaseous Hg passive air sampling (McLagan et al., 2016a). As such, we

recommend the use of the wind- and temperature-*adjusted SR*, but in the absence of available meteorological data, conclude that the *recalibrated SR* can be used with a high level of confidence, especially at sites not expected to have excessively high or low temperatures and wind speed. With substantially more data and with very minimal training of personnel, the precision of the instrument remains excellent ($3._6 \pm 3._0$ %). Furthermore, the PASs, with minimal upkeep under

some relatively harsh conditions, are considerably less prone than active instruments to issues resulting in data gaps. Overall, results are indicative of the PAS's potential as a tool for monitoring background gaseous Hg concentrations across a wide range of environmental conditions.

This study also attempted to address the exact nature of the analyte being taken up by the

sampler. At Mauna Loa, where overall GOM made up the greatest proportion of TGM, PAS results were significantly improved when using active GEM data over TGM data, which agrees with the hypothesis that GOM is removed by the diffusive barrier. Data at Alert were inconclusive as there were no significant differences with the PAS results using either active data for GEM or TGM for either the atmospheric Hg depletion event period or the whole dataset. While the Mauna Loa data

do suggest the PAS is taking up solely GEM, the same results were not apparent at Alert, and hence we cannot yet conclude with certainty that GEM is indeed the sole analyte sorbed by the





PASs. Furthermore, in all cases, the proportion of GOM (TGM minus GEM) in TGM measurements was close to the level of PAS uncertainty, which further reduces the strength of the conclusions that can be drawn. The deployment of samplers in controlled chambers with a point source of 530   GOM or isotopic analysis of the sorbed Hg may yield more definitive findings.

McLagan et al. (2016a) outlined three key rationales behind the development and use of a gaseous Hg PAS: (i) background concentration monitoring, especially at remote sites, (ii) measuring gaseous Hg gradients with high spatial resolution deployments, and (iii) personal exposure sampling. Results of this study indicate the PAS is a highly precise and accurate tool 535   that can complement and even replace existing monitoring techniques in certain circumstances across the three aforementioned rationales. Additionally, their small size, low cost, non-electrical operation and applicability across a range of conditions ascribe their versatility and with consideration may unlock a number of additional deployment scenarios that were not previously viable or even considered with only active monitoring instruments.

## Acknowledgements

We would like to sincerely thank all the site technicians and members of AMNet and ECCC-AMM Hg monitoring networks that assisted in the deployment and collection of the PASs and the retrieval and upkeep of active samplers. These individuals are: Dylan Nordin, Rob Tordon, Martin Pilote, Corrine Schiller, Helena Dryfhout-Clark, Kevin Rawlings, Melody Fraser, Matthew Hirsch, 545   Ronald Cole, Justin Chaffin, Andy Hale, Larry Scrapper, Nash Kobayashi, Da-Wei Lin, and Jin Sheng Chen. We also acknowledge funding from Strategic Project Grant no. 463265-14 by the Natural Sciences and Engineering Research Council of Canada (NSERC) and an NSERC Alexander Graham Bell Canada Graduate Scholarship. Alexandra Steffen acknowledges funding from the Northern Contaminants Program of Indigenous and Northern Affairs Canada for atmospheric Hg 550   monitoring at Alert and Little Fox Lake.

## Supplement

Supplementary information includes graphical, written, (linked) video standard operating procedures, all metadata, and active – passive concentration comparisons for each site. The nomenclature "Figure SX" (i.e. Fig. S2.1) refers to figures in the supplement.





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



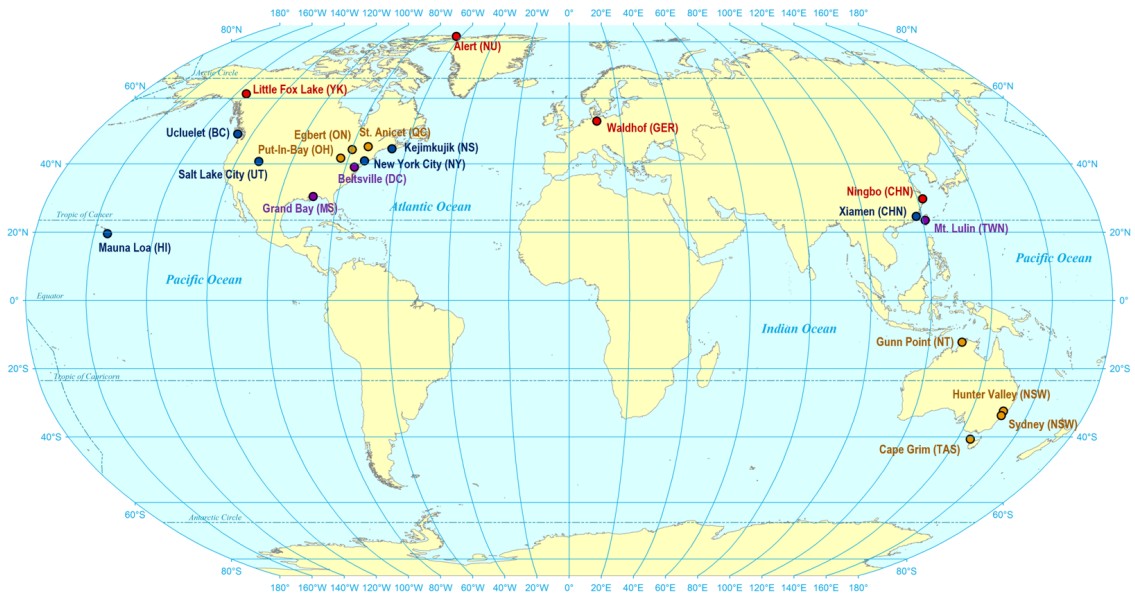


**Figure 1: Sampling sites for passive air sampler accuracy testing. Sampling sites are coloured by intensity of PAS deployments: high (monthly, seasonal, half yearly, and yearly deployments), medium (seasonal, half yearly, and yearly deployments), low (half yearly and yearly deployments), and very low (yearly deployments).**





**Figure 2: Uptake curves and individual deployments of passive air samplers across all 20 sampling locations. 0 ng points mark the beginning of deployments. All samples on the same line are for deployments that began at the same time. All axes are scaled the same. Deployments of the same colour cover equivalent deployments periods (i.e. orange is 7-12 month deployment at all sites).**



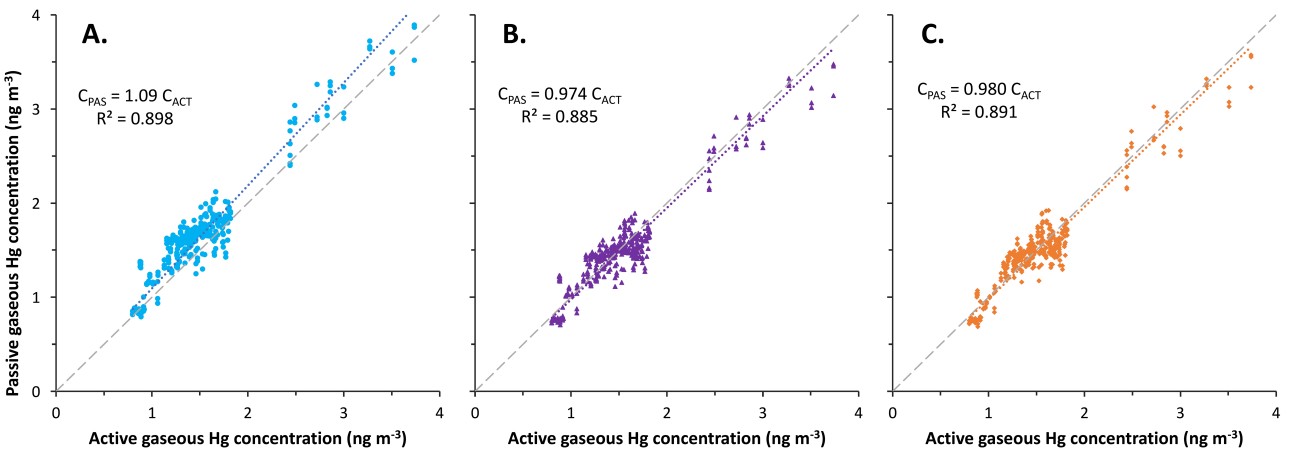

**Figure 3: Comparison of active (C$_{ACT}$; *x*-axis) and passive (C$_{PAS}$; *y*-axis) gaseous Hg concentrations derived from the *original sampling rate* (*SR*; Panel A), *recalibrated SR* (Panel B), and *adjusted SR* (Panel C). Dotted lines represent the trendline for each dataset. Grey dashed line is the 1:1 relationship.**



**Table 1: Mean passive and active air sampler data at each sampling location.**

**Part 1: Replicate relative standard deviation, uptake rate, original sampling rate (SR), recalibrated SR, and adjusted SR adjusted for the measured temperature and wind speed at each sampling site (Eq. 3).**

| Site | Intended deployments | Completed deployments | Deployments for comparison | Samplers for comparison ($n$) | Replicate RSD (%) | Uptake rate (ng day$^{-1}$) | Calculated SR (m$^3$ day$^{-1}$) | Temp./wind speed *adjusted* SR (m$^3$ day$^{-1}$) |
|---|---|---|---|---|---|---|---|---|
| Ningbo | 19 | 19 | 10 | 30 | $3._9 \pm 2._7$ | $0.35_7 \pm 0.04_8$ | $0.129_7 \pm 0.009_1$ | $0.134_6 \pm 0.003_1$ |
| Xiamen | 7 | 7 | 0 | 0 | $4._3 \pm 2._2$ | $0.38_6 \pm 0.03_6$ | - | - |
| Mt. Lulin | 1 | 1 | 1 | 3 | 1.2 | $0.202_8 \pm 0.002_5$ | $0.128_1 \pm 0.001_6$ | $0.137_3$ |
| Salt Lake City | 7 | 7 | 1 | 3 | $4._8 \pm 2._8$ | $0.234_2 \pm 0.015$ | $0.144_6 \pm 0.011$ | $0.150_3$ |
| Beltsville | 3 | 3 | 3 | 9 | $2.5 \pm 1.8$ | $0.195_8 \pm 0.010$ | $0.156_4 \pm 0.004_2$ | $0.133_2 \pm 0.002_8$ |
| Put-In-Bay | 3 | 3 | 3 | 9 | $1.2 \pm 0.4$ | $0.226_2 \pm 0.008_8$ | $0.141_4 \pm 0.002_9$ | $0.143_9 \pm 0.003_0$ |
| Grand Bay | 3 | 1 | 1 | 3 | 1.9 | $0.212_9 \pm 0.004_0$ | $0.159_6 \pm 0.003_0$ | $0.142_6$ |
| New York City | 7 | 7 | 7 | 21 | $3.9 \pm 1.8$ | $0.218_5 \pm 0.015$ | $0.123_7 \pm 0.006_3$ | $0.133_9 \pm 0.005_1$ |
| Mauna Loa | 7 | 7 | 7 | 21 | $2.8 \pm 1.9$ | $0.103_1 \pm 0.003_3$ | $0.119_6 \pm 0.006_2$ | $0.136_2 \pm 0.001_4$ |
| Kejimkujik | 7 | 7 | 7 | 20 | $3._2 \pm 2._9$ | $0.160_2 \pm 0.012$ | $0.136_1 \pm 0.006_3$ | $0.130_0 \pm 0.005_5$ |
| Little Fox Lake | 17 | 17 | 17 | 49 | $3._2 \pm 2._2$ | $0.197_2 \pm 0.016$ | $0.136_3 \pm 0.009_0$ | $0.130_5 \pm 0.007_2$ |
| Alert | 19 | 14 | 14 | 36 | $3._0 \pm 3._4$ | $0.20_0 \pm 0.04_8$ | $0.134_6 \pm 0.012$ | $0.133_4 \pm 0.005_5$ |
| Ucluelet | 7 | 7 | 7 | 21 | $2._9 \pm 2._3$ | $0.186_2 \pm 0.010$ | $0.144_4 \pm 0.006_3$ | $0.134_2 \pm 0.002_2$ |
| St. Anicet | 3 | 3 | 3 | 9 | $1.4 \pm 0.6$ | $0.196_8 \pm 0.003_2$ | $0.158_9 \pm 0.007_3$ | $0.137_3 \pm 0.005_5$ |
| Egbert | 3 | 3 | 3 | 9 | $1.5 \pm 0.2$ | $0.206_0 \pm 0.002_8$ | $0.145_5 \pm 0.004_2$ | $0.133_9 \pm 0.005_1$ |
| Waldhof | 17 | 17 | 17 | 46 | $6._9 \pm 4._8$ | $0.20_6 \pm 0.02_0$ | $0.124_0 \pm 0.013$ | $0.134_4 \pm 0.004_0$ |
| Hunter Valley | 3 | 1 | 0 | 0 | 3.8 | $0.155_4 \pm 0.005_4$ | - | - |
| Sydney | 3 | 2 | 0 | 0 | $3.2 \pm 1.2$ | $0.140_4 \pm 0.005_0$ | - | - |
| Cape Grim | 3 | 3 | 3 | 8 | $1.8 \pm 0.8$ | $0.162_9 \pm 0.003_0$ | $0.184_7 \pm 0.003_5$ | $0.157_2 \pm 0.002_0$ |
| Gunn Point | 3 | 3 | 3 | 9 | $0.7 \pm 0.3$ | $0.142_2 \pm 0.005_3$ | $0.147_3 \pm 0.003_6$ | $0.157_2 \pm 0.000_6$ |
| **TOTAL** | **142** | **132** | **107** | **306** | $\mathbf{3._6 \pm 3._0}$ | **-** | $\mathbf{0.135_4 \pm 0.016}$ | $\mathbf{0.135_4 \pm 0.007_7}$ |





**Table 1 continued:**

**Part 2: Gaseous Hg concentrations (conc.) and uncertainty (mean normalized error; MND) when calculated with three different sampling rates**

| Site | Active conc. (ng m$^{-3}$) | Original SR Passive conc. (ng m$^{-3}$) | Original SR Uncertainty MND (%) | Recalibrated SR Passive conc. (ng m$^{-3}$) | Recalibrated SR Uncertainty MND (%) | Adjusted SR Passive conc. (ng m$^{-3}$) | Adjusted SR Uncertainty MND (%) |
|---|---|---|---|---|---|---|---|
| Ningbo | 2.93 ± 0.44 | 3.13 ± 0.41 | 8.5 ± 5.9 | 2.80 ± 0.37 | 6.4 ± 4.5 | 2.81 ± 0.38 | 6.8 ± 4.8 |
| Xiamen | - | - | - | - | - | - | - |
| Mt. Lulin | 1.58 | 1.67 ± 0.02 | 5.9 ± 1.3 | 1.50 ± 0.02 | 5.3 ± 1.2 | 1.47 ± 0.02 | 6.7 ± 1.2 |
| Salt Lake City | 1.67 | 1.99 ± 0.15 | 19.4 ± 9.1 | 1.78 ± 0.14 | 8.3 ± 5.6 | 1.60 ± 0.12 | 5.2 ± 5.8 |
| Beltsville | 1.25 ± 0.05 | 1.62 ± 0.08 | 29.2 ± 3.4 | 1.45 ± 0.07 | 15.5 ± 3.1 | 1.47 ± 0.10 | 17.5 ± 4.6 |
| Put-In-Bay | 1.42 ± 0.03 | 1.66 ± 0.05 | 16.8 ± 2.4 | 1.49 ± 0.05 | 4.4 ± 2.1 | 1.40 ± 0.07 | 3.1 ± 1.4 |
| Grand Bay | 1.33 | 1.76 ± 0.03 | 31.8 ± 2.4 | 1.57 ± 0.03 | 17.9 ± 2.2 | 1.49 ± 0.03 | 11.9 ± 2.1 |
| New York City | 1.77 ± 0.05 | 1.81 ± 0.12 | 4.3 ± 3.4 | 1.62 ± 0.11 | 8.6 ± 4.6 | 1.64 ± 0.12 | 7.6 ± 4.6 |
| Mauna Loa | 0.86 ± 0.04 | 0.85 ± 0.03 | 4.3 ± 2.8 | 0.76 ± 0.02 | 12.0 ± 4.5 | 0.76 ± 0.03 | 12.2 ± 4.6 |
| Kejimkujik | 1.18 ± 0.10 | 1.32 ± 0.10 | 12.5 ± 5.2 | 1.18 ± 0.09 | 3.7 ± 2.7 | 1.24 ± 0.13 | 5.1 ± 3.5 |
| Little Fox Lake | 1.45 ± 0.05 | 1.63 ± 0.13 | 13.5 ± 5.6 | 1.46 ± 0.11 | 4.4 ± 5.0 | 1.51 ± 0.12 | 6.3 ± 4.4 |
| Alert | 1.39 ± 0.21 | 1.54 ± 0.26 | 12.9 ± 8.6 | 1.38 ± 0.23 | 6.8 ± 6.7 | 1.40 ± 0.20 | 7.0 ± 6.0 |
| Ucluelet | 1.29 ± 0.05 | 1.54 ± 0.08 | 19.3 ± 5.2 | 1.38 ± 0.07 | 7.4 ± 3.3 | 1.39 ± 0.08 | 8.1 ± 3.1 |
| St. Anicet | 1.21 ± 0.05 | 1.59 ± 0.02 | 34.4 ± 6.0 | 1.42 ± 0.02 | 17.4 ± 5.4 | 1.40 ± 0.06 | 15.4 ± 1.7 |
| Egbert | 1.41 ± 0.03 | 1.70 ± 0.02 | 20.2 ± 3.4 | 1.52 ± 0.02 | 7.5 ± 3.1 | 1.54 ± 0.06 | 8.7 ± 2.1 |
| Waldhof | 1.66 ± 0.08 | 1.70 ± 0.17 | 9.0 ± 6.6 | 1.52 ± 0.15 | 10.6 ± 7.3 | 1.53 ± 0.15 | 11.0 ± 6.8 |
| Hunter Valley | - | - | - | - | - | - | - |
| Sydney | - | - | - | - | - | - | - |
| Cape Grim | 0.88 ± 0.00 | 1.35 ± 0.02 | 52.6 ± 2.9 | 1.20 ± 0.02 | 36.4 ± 2.6 | 1.03 ± 0.02 | 17.4 ± 2.2 |
| Gunn Point | 0.96 ± 0.01 | 1.17 ± 0.04 | 21.7 ± 3.0 | 1.05 ± 0.04 | 8.8 ± 2.7 | 0.90 ± 0.03 | 6.3 ± 2.0 |
| **TOTAL** | **1.52 ± 0.47** | **1.54 ± 0.51** | **14.2 ± 10** | **-** | **8.8 ± 7.3** | **-** | **8.7 ± 5.7** |