# Peer review of "Global evaluation and calibration of a passive air sampler for gaseous mercury"

_Atmospheric Chemistry and Physics, 2017_

## Referee Comment (RC1) · Anonymous Referee #2 · 22 Feb 2018

A very interesting paper which provides a useful technical advance in an area of current importance to atmospheric monitoring.

---

## Referee Comment (RC2) · Anonymous Referee #3 · 22 Feb 2018

The present manuscript sounds as an extremely interesting study, useful for the scientific community involved in the global Hg monitoring in the atmosphere. Indeed this study accurately reports the results of a monitoring campaign comprising 20 sites worldwide, where active measurements of TGM (GEM+GOM) were performed simultaneously by validated analytical instruments.

A complete description of the PASs, as well as the sampling procedure and data analysis are reported, letting us to suppose that these easy-to-use devices can be a promising solution to implement the mercury monitoring network over the world.

On the other hand, a more precise and clear description about the Blanks PASs should be provided in order to better clarify the procedure of the "Blanks" exposure (reported within the Supplements), their composition and handling, time of exposure and the

meaning of their data tratment.

---

## Referee Comment (RC3) · Anonymous Referee #1 · 23 Feb 2018

This paper provides data from around the world collected using a GEM passive sampler. Based on previous work with GOM passive samplers Huang and Gustin 2015 I doubt that this system collects GOM. I appreciate the effort to get so many people to collaborate.

The paper is well written, and the figures and tables relevant. I think it would be really useful if the authors would make the sites as they have separated them out in the text as urban, rural, and high elevation with different symbols on Figure 3. I wonder also if it would be useful to make a graph that shows correlations of sampler uptake concentrations with Tekran concentrations that plot data based on the time resolution instead of lumping all into one figure?

The limitation of this method is the long time resolution and lack of collection of GOM

that is really the atmospheric form of most concern. The authors need to be honest about this. If there are short higher periods of exposure of GEM would the sampler resolve this in anyway given the very long sampling time? I also wonder about the activated carbon material. Is the uptake only surficial or can Hg penetrate into the interior given the design of the sampler? Some discussion of the past use of this sampling system for other gases should be mentioned (O3, nitrogen compounds) as well as any limitations.

Linear regression is not R2 it is r2 and these should be associated with a p-value. I also am not sure of the utility of this method overall. I would personally not promote this as a device that could be used for personal exposure sampling especially since GEM concentrations measured are so below the human health exposure limit. This certainly needs to be tested before being promoted.

Figure 1. I think it is misleading to present lines for 2 points and even 3 when the intercept starts with the blank. The blank has been subtracted from the sample so I am not sure if it is a relevant point. I think it might be more appropriate to just present data from each location with the same time resolution. This may better illustrate the different slopes. Just a thought.

---

## Editor Comment (EC1) · A. Dommergue (Editor) · 26 Feb 2018

The authors should state how the overall uncertainty budget for the measurements made by the samplers are expected to compare with the precision based uncertainties that they have calculated. i.e. is repeatability the dominant factor in the uncertainty or are their contributions that they may have missed or cannot easily estimate.

---

## Author Comment (AC1) · 16 Mar 2018

REVIEWER COMMENT: The present manuscript sounds as an extremely interesting study, useful for the scientific community involved in the global Hg monitoring in the atmosphere. Indeed, this study accurately reports the results of a monitoring campaign comprising 20 sites worldwide, where active measurements of TGM (GEM+GOM) were performed simultaneously by validated analytical instruments. A complete description of the PASs, as well as the sampling procedure and data analysis are reported, letting us to suppose that these easy-to-use devices can be a promising solution to implement the mercury monitoring network over the world.

RESPONSE: We appreciate the positive sentiments and support of the study.

[Figure]

REVIEWER COMMENT: On the other hand, a more precise and clear description about the Blanks PASs should be provided in order to better clarify the procedure of the "Blanks" exposure (reported within the Supplements), their composition and handling, time of exposure and the meaning of their data treatment.

RESPONSE: The exact procedure, handling and time of exposure for the field blanks is outlined in Section S1 of the supplementary information. In a revised version of the manuscript the following could be added to the main paper to better relay this information: Line 203-204: "Field blank samplers were used at each site and their composition/makeup is identical to regular samplers. Line 207-208: "Exact sampling procedures for field blanks are outlined in Section S1."

---

## Author Comment (AC4) · 6 Apr 2018

We appreciate the positive assessment of our manuscript.

---

## Author Response (AR1)

**Response to Anonymous Referee #1**

**REVIEWER COMMENT:** This paper provides data from around the world collected using a GEM passive sampler. Based on previous work with GOM passive samplers Huang and Gustin 2015 I doubt that this system collects GOM. I appreciate the effort to get so many people to collaborate.

**RESPONSE:** We appreciate the positive feedback and agree with the reviewer that the passive sampler is unlikely to take up GOM due to the diffusive barrier. We hope to confirm this with follow-up experiments in the near future.

**REVIEWER COMMENT:** The paper is well written, and the figures and tables relevant. I think it would be really useful if the authors would make the sites as they have separated them out in the text as urban, rural, and high elevation with different symbols on Figure 3.

**RESPONSE:** A version of Figure 3 that takes up the reviewer's suggestion is given here and has been included in the revised version of the manuscript.

[Figure]

In this figure the data are divided by color according to site type: red – urban sites; blue – rural sites; purple – high altitude sites; and yellow – northern/Arctic sites. The fitted relationships are for all data combined.

**REVIEWER COMMENT:** I wonder also if it would be useful to make a graph that shows correlations of sampler uptake concentrations with Tekran concentrations that plot data based on the time resolution instead of lumping all into one figure?

**RESPONSE:** Another version of Figure 3, which uses differently colored markers to represent data obtained over different deployment lengths, is given here:

[Figure]

In this figure blue, yellow, red and purple markers indicate 1, 3, 6 and 12 month deployments, respectively. (There were only two 9 month deployments.)

No significant effect of deployment length on the MND of samples for either the recalibrated SR (p = 0.082) or the adjusted SR (p = 0.298) was observed. Thus, neither the recalibrated nor adjusted SRs nor the uncertainty of the sampler is dependent upon the length of deployment at these background concentrations over deployments from 1 to 12 months in length. This figure has been added to the supplementary information of the revised manuscript.

**REVIEWER COMMENT:** The limitation of this method is the long time resolution and lack of collection of GOM that is really the atmospheric form of most concern. The authors need to be honest about this. If there are short higher periods of exposure of GEM would the sampler resolve this in anyway given the very long sampling time?

**RESPONSE:** We agree that a passive air sampler cannot achieve the same temporal resolution as an active instrument. It is, however, noteworthy to point out that while the shortest deployment in the current study was one month, the sampler can take up amounts of Hg that are sufficient for reliable quantification in much shorter time periods. We have previously estimated that at atmospheric background levels (1.5 ng/m$^3$), a temporal resolution as short as 5 days is achievable (McLagan et al. 2016). At higher concentrations much shorter deployment periods are possible. For example, at concentrations of 10 ng/m$^3$, 100 ng/m$^3$ and 1000 ng/m$^3$, the shortest PAS deployment times to yield amounts above the MQL (0.86 ng of Hg) are estimated to be ~1 day, ~2 hours and ~20 minutes. As accuracy and precision may deteriorate close to the MQL, it is advisable to sample somewhat longer than those minimums.

We do not agree that the "lack of collection of GOM" is a limitation of the sampler. In fact, the sampler was specifically developed to monitor gaseous elemental mercury (GEM). The Minimata Convention (UNEP 2013) and recent papers on the state of atmospheric mercury science have stated the need for improved monitoring of GEM/total gaseous Hg (TGM) due to the limited and biased spatial coverage of current monitoring sites (e.g. Pirrone et al. 2013). Atmospheric models predict spatially highly variable GEM/TGM concentrations in some of the areas with the poorest GEM/TGM measurement coverage (Travnikov et al. 2017). Dry deposition of GEM appears be a much more important pathway for atmospheric Hg deposition than previously thought (Obrist et al. 2017). Accordingly, we do not necessarily share the opinion, that GOM is "the atmospheric form of most concern". GEM is by far the most prevalent atmospheric mercury species and contributes most to its global dispersion.

We take exception to the insinuation that we are somehow dishonest about the limitations of the PAS. The limited temporal resolution of the PAS has been spelled out on lines 286-288 ("The time-averaged nature of the concentrations measured by the PASs conceals much of the variability that generally occurs at shorter time resolution.") We are equally forthright about our current inability to establish with certainty whether GOM is being taken up by the PAS or not. On lines 526-529 we wrote: "we cannot yet conclude with certainty that GEM is indeed the sole analyte sorbed by the PASs. Furthermore, in all cases, the proportion of GOM (TGM minus GEM) in TGM measurements was close to the level of PAS uncertainty, which further reduces the strength of the conclusions that can be drawn."

**REVIEWER COMMENT:** I also wonder about the activated carbon material. Is the uptake only surficial or can Hg penetrate into the interior given the design of the sampler? Some discussion of the past use of this sampling system for other gases should be mentioned (O3, nitrogen compounds) as well as any limitations.

**RESPONSE:** The mercury taken up by the carbon sorbent is quantified by combusting all of the carbon in a total mercury analyzer and not by desorbing it thermally. Therefore, while it may be of academic interest to explore the nature of the uptake of Hg in the sorbent, it has no bearing on the performance of the sampler. It also means that a comparison with other vapors that are analyzed differently is unlikely to provide useful insights.

**REVIEWER COMMENT:** Linear regression is not R2 it is r2 and these should be associated with a p-value.

**RESPONSE:** Both $r^2$ an $R^2$ are in common usage to designate the coefficient of determination. The revised version of Figure 3 includes the p-values associated with the regressions.

**REVIEWER COMMENT:** I also am not sure of the utility of this method overall.

**RESPONSE:** We hope that we eventually will be able to convince this reviewer of the tremendous potential that the passive air sampler for mercury holds. We have already applied the sampler in a number of studies (e.g. characterization and quantification of mercury from area sources, identification of unknown mercury sources), which will be published in the near future. The potential applications are varied and numerous.

**REVIEWER COMMENT:** I would personally not promote this as a device that could be used for personal exposure sampling especially since GEM concentrations measured are so below the human health exposure limit. This certainly needs to be tested before being promoted.

**RESPONSE:** While the maximum gaseous mercury concentrations measured in the current study are below 4 ng/m$^3$, we have already applied the sampler to record concentrations that are more than four orders of magnitude higher. While we think that the sampler has the potential to be used for monitoring mercury in workplace atmospheres and for personal exposure sampling, we agree with the reviewer that more testing is required, before the PAS can be confidently applied for this purpose.

**REVIEWER COMMENT:** Figure 1. I think it is misleading to present lines for 2 points and even 3 when the intercept starts with the blank. The blank has been subtracted from the sample so I am not sure if it is a relevant point. I think it might be more appropriate to just present data from each location with the same time resolution. This may better illustrate the different slopes. Just a thought.

**RESPONSE:** We assume the reviewer refers to Figure 2. We do not agree with the reviewer's suggestions. Yes, the blank has been subtracted from the samples, hence each uptake period starts at the origin and not at the blank level. The slope of each curve, even those with only two or three data points, corresponds to an uptake rate (ng/day) over the deployment period and conveys valuable information on whether the sampler remains in the linear uptake phase. The passive data from each time period are given in the current figure format and the different slopes are also apparent, as the scales of each graph are the same.

**Response to Anonymous Referee #3**

**REVIEWER COMMENT:** The present manuscript sounds as an extremely interesting study, useful for the scientific community involved in the global Hg monitoring in the atmosphere. Indeed, this study accurately reports the results of a monitoring campaign comprising 20 sites worldwide, where active measurements of TGM (GEM+GOM) were performed simultaneously by validated analytical instruments. A complete description of the PASs, as well as the sampling procedure and data analysis are reported, letting us to suppose that these easy-to-use devices can be a promising solution to implement the mercury monitoring network over the world.

**RESPONSE:** We appreciate the positive sentiments and support of the study.

**REVIEWER COMMENT:** On the other hand, a more precise and clear description about the Blanks PASs should be provided in order to better clarify the procedure of the "Blanks" exposure (reported within the Supplements), their composition and handling, time of exposure and the meaning of their data treatment.

**RESPONSE:** The exact procedure, handling and time of exposure for the field blanks is outlined in Section S1 of the supplementary information. In the revised version of the manuscript the following sentence has been added to section 2.4.1:

"The mean concentrations in field blanks, i.e. unexposed samplers undergoing the same transport as regular PAS (for detail see Section S1), are listed in Table S2.4 for each sampling site."

**Response to Editor**

**EDITOR COMMENT:** The authors should state how the overall uncertainty budget for the measurements made by the samplers are expected to compare with the precision based uncertainties that they have calculated, i.e. is repeatability the dominant factor in the uncertainty or are their contributions that they may have missed or cannot easily estimate.

**RESPONSE:**

The precision estimate that we report (precision: $3_{.6} \pm 3_{.0}$ %) is calculated as the average of the standard deviation of the results for replicated deployments of the passive air sampler (PAS). As such it is a measure of random error only. It cannot capture systematic bias.

Quantifying the systematic error would require knowledge of the true gaseous concentration of mercury during a PAS's deployment. Because that concentration is not known, we instead compare the concentrations obtained with the PAS with values obtained with the state-of-the-art measurement technique (Tekran). The Tekran systems do not provide the true value, because (1) they are subject to random and systematic error themselves and (2) only in some cases did they succeed in measuring the concentration continuously during the entire deployment period of the PAS.

Nevertheless we use the discrepancy between the concentration obtained with the PAS and the Tekran (mean normalised difference or MND of $8_{.7} \pm 5_{.7}$ %, when the "best" SR is applied) as an estimate of the potential systematic uncertainty of the PAS. On the one hand, this MND overestimates the uncertainty of the PAS by attributing all of the discrepancy to it, even though part of the discrepancy is surely attributable to the Tekran. On the other hand, we may underestimate the uncertainty of the PAS, because we use the same TEKRAN data for the calibration of the sampling rate and the calculation of the MND. If all of the Tekrans in our study were biased similarly low (or high), this bias would be "inherited" by the PAS and this uncertainty would not be apparent in the MND.

If, however, some Tekran data are biased high and some are biased low, this would not lead to an underestimation of the uncertainty of the PAS, because we derive a single sampling rate from all data and apply it (after adjustment for wind speed and temperature) to all sites, i.e. we do not use site-specific sampling rates for calibration AND evaluation.

So, in summary, it is likely that there are uncertainties in the PAS-derived values that are not captured in the precision based estimate, i.e. the uncertainty is likely higher than 3.6 %. It is difficult to really quantify this additional uncertainty, but based on the current study we judge the overall uncertainty of the PAS to be on average smaller than 8.7%. In the

revised version, the section with the title "**3.4 Placing the PAS performance into context**" has been expanded and edited to reflect the above discussion.

$$\quad SR_{adj} = SR_{cal} + \left(T_{exp} - 9.89\ °\text{C}\right) \cdot 0.0009\ \frac{m^3}{day\ °\text{
[revised manuscript text omitted]

Additionally, deployment length had no significant effect on the MND of samples for either the *recalibrated SR* ($p$ = 0.082) or the *adjusted SR* ($p$ = 0.298). Thus, neither the *recalibrated* nor *adjusted SRs* nor the uncertainty of the sampler is dependent upon the length of deployment at background concentrations over deployments lasting 1 to 12 months (Fig. S3.1). Nine month deployments were not considered in this analysis as there were only two deployments of this length at Alert and Little Fox Lake.

**3.4 Placing the PAS performance into context**

It is important to acknowledge that this is not a fully independent evaluation of the performance of the sampler, as the same data were used in the derivation of the *recalibrated SR* and in the determination of the air concentrations from the PAS. However, we stress that data from all sites were used to derive a single *recalibrated SR* that was used for all sites, i.e. the fitting involved was not site-specific. For example, the Little Fox Lake site contributed the most data points to the recalibration ($n$ = 49), but those data only represented 14 % of the whole data set.

Our precision estimate ($3._6 \pm 3._0$ %) calculated as the average of the standard deviation of the results for replicated deployments of the passive air sampler, is a measure of random error only. Quantifying the systematic error would require knowledge of the true gaseous concentration of mercury during a PAS's deployment. Because that concentration is not known, we instead use the MND between the concentration obtained with the PAS and the Tekran systems as an estimate of the potential systematic uncertainty of the PAS ($8._7 \pm 5._7$ % when the adjusted *SR* is applied). On the one hand, this MND overestimates the uncertainty of the PAS by attributing all of the discrepancy to it, even though part of the discrepancy is surely attributable to the Tekran. By sampling with two collocated Tekran 2537A instruments, Temme et al. (2007) estimated a measurement uncertainty of 8.8 %. Aspmo et al. (2005) describe the uncertainty of the same system in the range of 5-10 %. Based on a review of several inter-comparisons involving Tekran 2537 instruments, Slemr et al. (2015) estimated a systematic uncertainty of ≈10 %, but warn this can expand up to 20 % in extreme cases. Even if the Tekran systems were to yield true values of the concentrations, the MND can be partly attributed to data gaps in the active instrument measurements (see Table S2.2) that if successfully analyzed would have altered the active concentration at that site in some way.

On the other hand, there is also the possibility that the MND underestimates the systematic error of the PAS, namely if most or all of the Tekran systems in our study were biased similarly low (or high). Because the same TEKRAN data are used for both the sampling rate calibration and the calculation of the MND, this bias would be "inherited" by the PAS and therefore not be apparent in the MND. If, however, some Tekran data are biased high and some are biased low, this would be unlikely to lead to an underestimation of the uncertainty of the PAS.

In summary, we judge the overall uncertainty of the PAS to be higher than the precision based uncertainty ($3_{.6} \pm 3_{.0}$ %), because of the potential for systematic bias, but also not higher on average than the MND obtained with the *adjusted SR* ($8_{.7} \pm 5_{.7}$ %), because that discrepancy is not solely attributable to systematic bias by the PAS. Even this conservative assessment of PAS accuracy is in line with active instruments' uncertainty and qualifies the device as appropriate for background monitoring of gaseous Hg.

The performance of the PAS using either the *recalibrated* or *adjusted SR* represents a substantial improvement over all existing gaseous Hg PAS designs to date, especially those with sufficiently low detection limits to monitor background concentrations (as summarized in a review on gaseous Hg PASs by McLagan et al., 2016a). While the accuracy-based uncertainty of the 3M PAS by McCammon and Woodfin (McCammon and Woodfin, 1977) was similar (8 ± 7 %), this device was only tested in the range of 25,000 – 300,000 ng m$^{-3}$ gaseous Hg concentrations over an 8-hour period, making it unsuitable for background monitoring. Of the PASs that have sufficiently low detection limits to monitor background concentrations the lowest overall uncertainties were 19 ± 14 (Huang et al., 2012) and 22 ± 15 % (Guo et al., 2014; Zhang et al., 2012). Other designs had uncertainties greater than 30 % (Brown et al., 2012; Nishikawa et al., 1999), reported only replicate precision (Brumbaugh et al., 2000; Skov et al., 2007), or reported no uncertainty estimate at all (Gustin et al., 2011).

**3.5 Site specific analysis**

Plots comparing active instrument derived gaseous Hg concentrations with passive concentrations determined using the *original*, *recalibrated*, and *adjusted SRs* for each sampling site are presented in Section S4 (Fig. S4.1 – S4.17). The data in Fig. 3 are colour coded by site categorization (urban, rural, altitude, northern/Arctic).

Frank Wania 2018-4-10 15:15

Frank Wania 2018-4-10 15:15

Frank Wania 2018-4-10 15:15

Frank Wania 2018-4-10 15:15

Frank Wania 2018-4-10 15:15

**3.5.1 Urban Sites**

[revised manuscript text omitted]